REPLICATION STUDY

# Replication Study: BET bromodomain inhibition as a therapeutic strategy to target c-Myc

**Fraser Aird, Irawati Kandela, Christine Mantis, Reproducibility Project: Cancer Biology***

Developmental Therapeutics Core, Northwestern University, Evanston, United States

**Abstract** In 2015, as part of the Reproducibility Project: Cancer Biology, we published a Registered Report (Kandela et al., 2015) that described how we intended to replicate selected experiments from the paper "BET bromodomain inhibition as a therapeutic strategy to target c-Myc" (Delmore et al., 2011). Here we report the results of those experiments. We found that treatment of human multiple myeloma (MM) cells with the small-molecular inhibitor of BET bromodomains, (+)-JQ1, selectively downregulated *MYC* transcription, which is similar to what was reported in the original study (Figure 3B; Delmore et al., 2011). Efficacy of (+)-JQ1 was evaluated in an orthotopically xenografted model of MM. Overall survival was increased in (+)-JQ1 treated mice compared to vehicle control, similar to the original study (Figure 7E; Delmore et al., 2011). Tumor burden, as determined by bioluminescence, was decreased in (+)-JQ1 treated mice compared to vehicle control; however, while the effect was in the same direction as the original study (Figure 7C-D; Delmore et al., 2011), it was not statistically significant. The opportunity to detect a statistically significant difference was limited though, due to the higher rate of early death in the control group, and increased overall survival in (+)-JQ1 treated mice before the pre-specified tumor burden analysis endpoint. Additionally, we evaluated the (−)-JQ1 enantiomer that is structurally incapable of inhibiting BET bromodomains, which resulted in a minimal impact on *MYC* transcription, but did not result in a statistically significant difference in tumor burden or survival distributions compared to treatment with (+)-JQ1. Finally, we report meta-analyses for each result.

*For correspondence: tim@cos.io; nicole@scienceexchange.com

Group author details:
Reproducibility Project: Cancer Biology See page 12

## Introduction

The Reproducibility Project: Cancer Biology (RP:CB) is a collaboration between the Center for Open Science and Science Exchange that seeks to address concerns about reproducibility in scientific research by conducting replications of selected experiments from a number of high-profile papers in the field of cancer biology (*Errington et al., 2014*). For each of these papers a Registered Report detailing the proposed experimental designs and protocols for the replications was peer reviewed and published prior to data collection. The present paper is a Replication Study that reports the results of the replication experiments detailed in the Registered Report (*Kandela et al., 2015*), for a paper by Delmore et al., and uses a number of approaches to compare the outcomes of the original experiments and the replications.

In 2011, Delmore et al. demonstrated that inhibition of BET bromodomains with a selective small-molecule, (+)-JQ1, down-regulated the c-Myc transcriptional signaling network and reduced tumor burden and prolonged survival *in vivo* indicating that targeting of BET bromodomains is an effective strategy to modulate c-Myc function in multiple myeloma (MM). Time-dependent downregulation of *MYC* was observed in a human MM cell line (MM.1S) treated with (+)-JQ1, in agreement with other

examined MM cell lines (*Delmore et al., 2011*). Using a bioluminescent MM xenograft model (MM.1S-luc) daily treatment with (+)-JQ1 resulted in a statistically significant decrease in tumor burden and, importantly, increased overall survival compared to vehicle control treated animals.

The Registered Report for the paper by Delmore et al. described the experiments to be replicated (Figures 3B and 7C–E), and summarized the current evidence for these findings (*Kandela et al., 2015*). Since that publication there have been additional studies examining the therapeutic strategy of targeting BET bromodomains in other types of cancer. This includes reports of antitumor effects using BET bromodomain inhibitors in MM(*Chaidos et al., 2014* , *Siu et al., 2016*), ovarian cancer (*Zhang et al., 2016*), gastric cancer (*Montenegro et al., 2014*), childhood sarcoma (*Bid et al., 2016*), and triple negative breast cancer (*da Motta et al., 2016*; *Shu et al., 2016*). Acquired resistance to BET inhibitors have also been reported (*Fong et al., 2015*; *Kumar et al., 2015*; *Rathert et al., 2015*), with recent studies suggesting combinatorial drug treatment to overcome resistance mechanisms (*Asangani et al., 2016*; *Kurimchak et al., 2016*; *Yao et al., 2015*). In addition to efficacy, the nonclinical safety of BET inhibition has also been examined. In mesenchymal stem cells, (+)-JQ1 was reported to induce cell cycle arrest and downregulation of genes involved in self-renewal, mitosis, and DNA replication (*Alghamdi et al., 2016*), while mice treated with (+)-JQ1 at an efficacious dose resulted in lymphoid and hematopoietic toxicity (*Lee et al., 2016*). Currently, several BET bromodomain inhibitors, with slight variation in mechanism, are in clinical trials for patients with various hematologic and solid malignancies (*Chaidos et al., 2015*; *French, 2016*; *Wadhwa and Nicolaides, 2016*). Early results from a phase one study to establish the recommended dose of the OTX015/MK-8628 BET inhibitor in hematologic malignancies reported the drug was tolerated; however, thrombocytopenia was a common toxic effect observed (*Amorim et al., 2016*). In four patients with advanced stage NUT midline carcinoma, with confirmed BRD4-NUT fusions, early clinical benefit was reported for two, with a third achieving disease stabilization after treatment with OTX015/MK-8628 (*Stathis et al., 2016*).

The outcome measures reported in this Replication Study will be aggregated with those from the other Replication Studies to create a dataset that will be examined to provide evidence about reproducibility of cancer biology research, and to identify factors that influence reproducibility more generally.

## Results and discussion

### Evaluation of *MYC* expression in JQ1-treated MM.1S-luc Cells

We sought to independently replicate an experiment analyzing the expression of endogenous *MYC* during pharmacological inhibition of BET bromodomains with (+)-JQ1. This experiment is similar to what was reported in Figure 3B (*Delmore et al., 2011*) and assesses the levels of *MYC* by quantitative reverse transcription polymerase chain reaction (qRT-PCR) in a human MM cell line stabling expressing luciferase (MM.1S-luc) (*Mitsiades et al., 2004*). While the original study included a time course treatment with assessments at 0 hr, 0.5 hr, 1 hr, 4 hr, and 8 hr, the replication was restricted to the early- (0 hr and 1 hr) and late-treatment (8 hr) time points. Additionally, the replication experiment was extended to include additional control conditions to monitor *MYC* expression with the inactive (−)-JQ1 enantiomer (*Filippakopoulos et al., 2010*) and vehicle control at the same time points. We found that during the time course of cells treated with (+)-JQ1, *MYC* expression was inhibited by 0.146 times [n = 5, *SD* = 0.031] at 1 hr and 0.126 times [n = 5, *SD* = 0.036] at 8 hr following treatment compared to the amount of *MYC* expressed in cells at 0 hr [n = 5, M = 1.00, *SD* = 0.070] (*Figure 1*). This compares to the original study, which reported an estimated relative *MYC* expression of ~0.069 times and ~0.088 times for cells treated with (+)-JQ1 at 1 hr and 8 hr, respectively (*Delmore et al., 2011*). In MM.1S-luc cells treated with (−)-JQ1 or vehicle, the average relative *MYC* expression remained visually largely unchanged following treatment, indicating selectivity of *MYC* down-regulation through inhibition of BET bromodomains.

There are multiple approaches that could be taken to explore this data; however, to provide a direct comparison to the original analysis, we are reporting the analysis specified *a priori* in the Registered Report (*Kandela et al., 2015*). The mixed-design analysis of variance (ANOVA) result was statistically significant for all effects. A further analysis of the simple main effects for each treatment type revealed a statistically significant effect with (+)-JQ1 treatment ($F(2,8) = 777.5$, $p=6.86 \times 10^{-10}$),

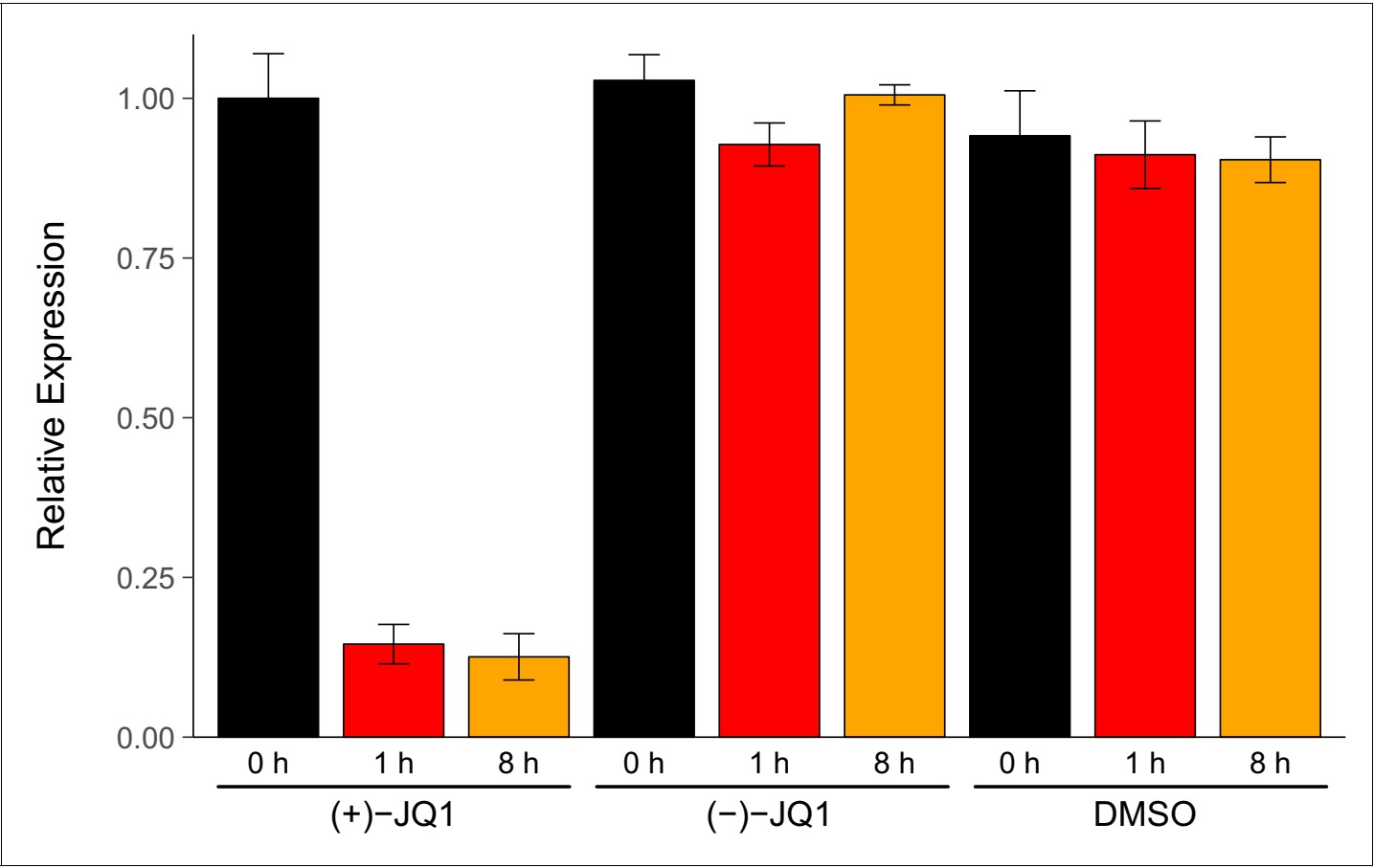

**Figure 1.** *MYC* expression in JQ1-treated MM.1S-luc cells. MM.1S-luc cells were treated with 500 nM (+)-JQ1, 500 nM (−)-JQ1, or an equivalent volume of DMSO. Total RNA was isolated at 0 hr, 1 hr, and 8 hr after treatment and qRT-PCR analysis was performed to detect *MYC* and *GAPDH* levels. Relative expression (*MYC/GAPDH*) is presented for each time point and condition normalized to (+)-JQ1 treated cells at 0 hr. Means reported and error bars represent s.d. from five independent biological repeats. Mixed-design analysis of variance (ANOVA) with time (0 hr, 1 hr, and 8 hr) as the within-subjects factor and treatment ((+)-JQ1, (−)-JQ1, or vehicle) as the between-subjects factor; interaction effect: $F_{(4,24)} = 268.9$, $p=1.49 \times 10^{-19}$, treatment main effect: $F_{(2,12)} = 393.5$, $p=1.15 \times 10^{-11}$, time main effect: $F_{(2, 24)} = 368.0$, $p=9.84 \times 10^{-19}$. Planned paired *t*-test of MM.1S-luc cells harvested 8 hr after (+)-JQ1 treatment compared to cells 0 hr after (+)-JQ1 treatment; $t(4) = 38.92$, uncorrected $p=2.60 \times 10^{-6}$, *a priori* Bonferroni adjusted significance threshold = 0.025; (Bonferroni corrected $p=5.21 \times 10^{-6}$). Planned paired *t*-test of MM.1S-luc cells harvested 1 hr after (+)-JQ1 treatment compared to cells 0 hr after (+)-JQ1 treatment; $t(4) = 25.10$, uncorrected $p=1.50 \times 10^{-5}$, *a priori* Bonferroni adjusted significance threshold = 0.025; (Bonferroni corrected $p=2.99 \times 10^{-5}$). Additional details for this experiment can be found at https://osf.io/9swnx/.

indicating that the mean *MYC* expression was different among the three time points evaluated. As outlined *a priori* in the Registered Report (*Kandela et al., 2015*), two planned paired *t*-tests were conducted as follow-up tests using the Bonferroni correction to adjust for multiple comparisons, making the significance threshold 0.025. The comparison of MM.1S-luc cells harvested 8 hr after (+)-JQ1 treatment compared to cells 0 hr after (+)-JQ1 treatment was statistically significant ($t(4) = 38.92$, uncorrected $p=2.60 \times 10^{-6}$, corrected $p=5.21 \times 10^{-6}$), as well as the comparison of MM.1S-luc cells harvested 1 hr after (+)-JQ1 treatment compared to cells 0 hr after (+)-JQ1 treatment ($t(4) = 25.10$, uncorrected $p=1.50 \times 10^{-5}$, corrected $p=2.99 \times 10^{-5}$). As expected from the data presented in *Figure 1*, examination of the simple main effect for vehicle control treatment was not statistically significant ($F(2,8) = 1.05$, $p=0.394$). Interestingly, though, treatment with (−)-JQ1 yielded a statistically significant effect ($F(2,8) = 19.40$, $p=0.00085$). While this indicates that the null hypothesis that there is no difference among the different time points can be rejected, the magnitude of the effect size in (−)-JQ1 treated cells, $\eta_G^2 = 0.698$, 90% CI [0.444, 0.882], is considerably smaller than (+)-JQ1 treated cells, $\eta_G^2 = 0.989$, 90% CI [0.980, 0.996]. Furthermore, *MYC* expression at 0 hr [n = 5, M = 1.028, SD = 0.040], 1 hr [n = 5, M = 0.928, SD = 0.034], and 8 hr [n = 5, M = 1.005,

*SD* = 0.016] after treatment with (−)-JQ1 is in stark contrast to what was observed when MM.1S-luc cells were treated with (+)-JQ1 (*Figure 1*).

In addition to our confirmatory analyses described above, the data were also explored as outlined in the Registered Report (*Kandela et al., 2015*). Although this was a planned analysis, it is exploratory because it differs from the methods used in the original study by analyzing the data as unpaired instead of paired. All of the effects of the two-way between-subjects ANOVA were statistically significant (interaction effect, $F(4,36)$ = 175.9, $p=4.28 \times 10^{-23}$; treatment main effect, $F(2,36)$ = 665.9, p=3.68x$10^{-29}$; time main effect, $F(2,36)$ = 240.6, $p=1.47 \times 10^{-21}$). The planned contrast of MM.1S-luc cells harvested 8 hr after (+)-JQ1 treatment and cells 0 hr after (+)-JQ1 treatment was statistically significant ($t(36)$ = 29.27, $p=1.06 \times 10^{-26}$, Cohen's $d$ = 15.82, 95% CI [8.14, 23.51]). Similarly, the planned contrast of MM.1S-luc cells harvested 1 hr after (+)-JQ1 treatment compared to cells 0 hr after (+)-JQ1 treatment also yielded statistically significant results ($t(36)$ = 29.96, $p=4.80 \times 10^{-26}$, Cohen's $d$ = 15.71, 95% CI [8.08, 23.35]). These results are in agreement with the mixed-design ANOVA and planned paired *t*-tests.

## Testing the efficacy of JQ1 treatment in mice harboring bioluminescent MM lesions

In order to test the effectiveness of (+)-JQ1 as a therapeutic strategy to target c-Myc, we sought to replicate experiments similar to what was reported in Figure 7C–E (*Delmore et al., 2011*). This included evaluation of tumor burden in mice orthotopically xenografted, as well as analysis of the overall rate of survival. Similar to the *in vitro* experiment, the inactive (−)-JQ1 enantiomer, which was not included in the original study, was added as an additional control to evaluate the specificity of the approach. Female SCID-beige mice were inoculated intravenously with a previously characterized bioluminescent MM model (MM.1S-luc) that recapitulates the clinical disease and allows for *in vivo* monitoring of tumor burden (*Mitsiades et al., 2004*). A pilot study was conducted to determine the appropriate timescale of tumor progression during this replication attempt after an initial attempt to engraft mice resulted in a longer delay from inoculation to detection than expected and reported in the original study. During the pilot study, mice were inoculated and monitored for tumor burden 8 days and 13 days after inoculation as described in the Registered Report (*Kandela et al., 2015*). The mice continued to be monitored three times a week until 57 days after inoculation when disease symptoms (e.g. significant weight loss and hind limb paralysis) associated with this xenograft model were detected in multiple mice. The baseline that was observed in the original study eight days after inoculation ($\sim 2 \times 10^6$ bioluminescence) was not observed until day 27 post-inoculation in the pilot study. Interestingly, other studies that utilized almost the same model (strain, sex, and age of mice as well as mode of injection) have also reported variable times to detectable engraftment. Tai and colleagues did not report baseline signals until two weeks after inoculation, granted $5 \times 10^6$ cells were injected opposed to the $2 \times 10^6$ cells injected in the original study and this replication attempt (*Tai et al., 2014*). Similarly, Zhang and colleagues did not observe this level of baseline bioluminescence until 2–3 weeks when injecting $3 \times 10^6$ cells (*Zhang et al., 2014*). Additionally, two other studies reported variable times to detectable engraftment, however these studies utilized different mouse strains and sex. When male SCID/NOD mice were injected with $5 \times 10^6$ cells, a baseline signal was not reported until two weeks after inoculation (*Mitsiades et al., 2004*), while Azab and colleagues, which inoculated $2 \times 10^6$ MM.1S-luc cells, reported that it took 3–4 weeks for sufficient tumor progression to occur and be detected (*Azab et al., 2009*).

Following the modified time frame, mice were inoculated with $2 \times 10^6$ MM.1S-luc cells and closely monitored until the desired baseline bioluminescence was achieved, which occurred on day 30 post-inoculation. After a further measurement five days later, all mice with established disease, defined as lesions diffusely engrafted in the skeleton with an increase in bioluminescence between days 30 and 35, were randomized into three cohorts. Following daily intraperitoneal injection of vehicle control, 50 mg/kg (+)-JQ1, or 50 mg/kg (−)-JQ1 for 36 days we assessed the overall survival rate of the animals (*Figure 2*). Mice treated with vehicle control achieved a median survival of 24.5 days during the treatment period [n = 12], which was lengthened during (+)-JQ1 treatment. The median survival of the (+)-JQ1 treated group could not be determined since more than half of the animals survived to the pre-specified study endpoint of 36 days. The median survival of mice treated with (−)-JQ1 [n = 12] was 29 days. This compares to the original study which achieved a median survival of 22 days with vehicle control [n = 10] that was prolonged to 35 days with (+)-JQ1 treatment [n = 9].

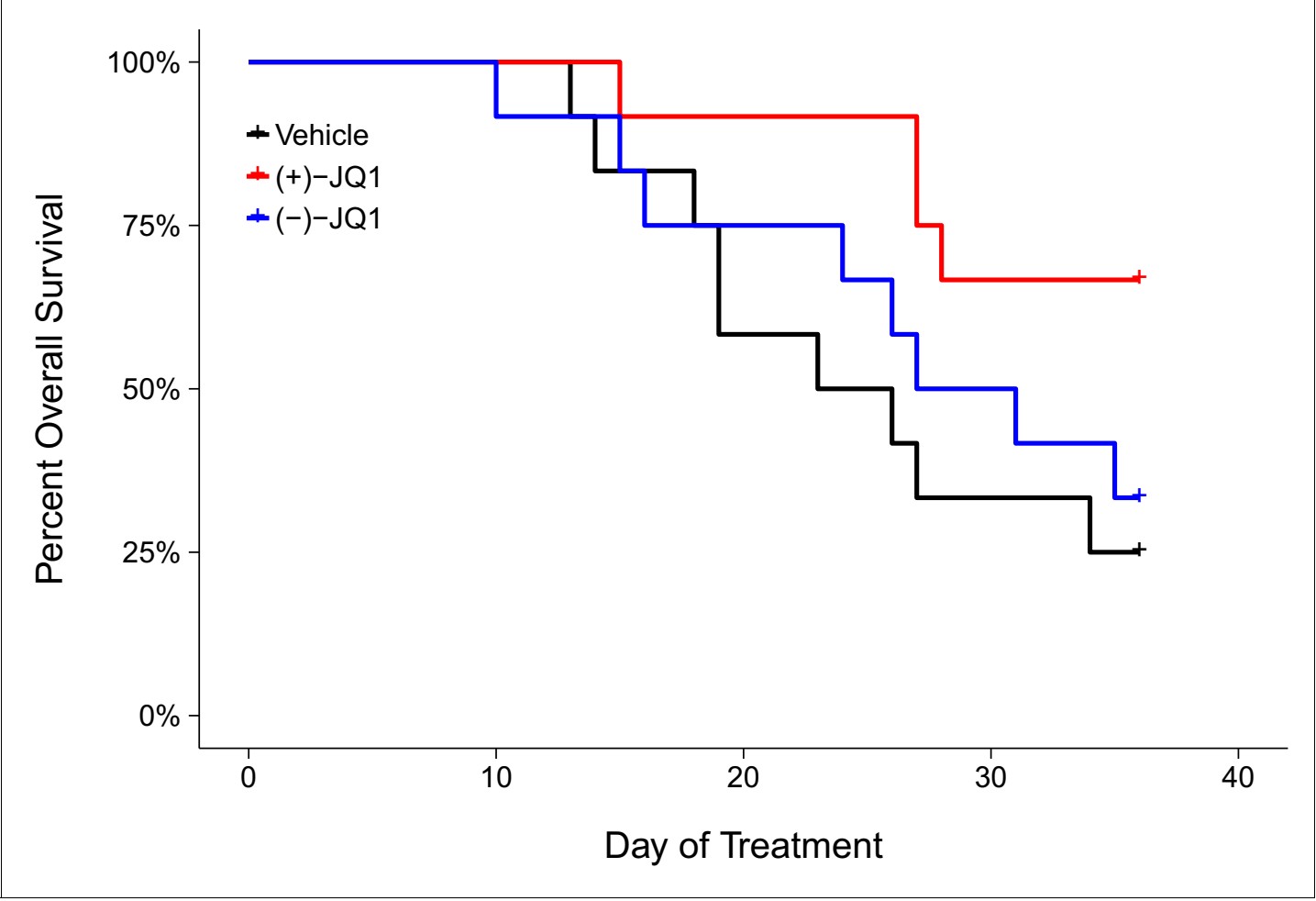

**Figure 2.** Overall survival in JQ1-treated MM.1S-luc orthotopic xenograft model. Kaplan-Meier plot of overall survival during the pre-specified study period of 36 days. Female SCID-beige mice with established MM.1S-luc orthotopic xenografts were randomized to daily IP injections of 50 mg/kg (+)-JQ1, 50 mg/kg (−)-JQ1, or vehicle control. Number of mice: n = 12 for each group. Log-rank (Mantel-Cox) test of (+)-JQ1 treatment compared to vehicle control; uncorrected $p=0.024$, *a priori* Bonferroni adjusted alpha level = 0.025; (Bonferroni corrected $p=0.047$). Log-rank (Mantel-Cox) test of (+)-JQ1 compared to (−)-JQ1; uncorrected $p=0.093$, *a priori* Bonferroni adjusted alpha level = 0.025; (Bonferroni corrected $p=0.187$). Additional details for this experiment can be found at https://osf.io/pnvtd/.

Furthermore, the median survival rates for vehicle treated mice are in agreement with control conditions in other studies reporting survival during treatment with this xenograft model (*Mitsiades et al., 2004*; *Tai et al., 2014*). To compare the survival distributions a log-rank (Mantel-Cox) test was performed. As outlined in the Registered Report (*Kandela et al., 2015*), we planned to conduct two comparisons using the Bonferroni correction to adjust for multiple comparisons making the significance threshold 0.025. The planned comparison between (+)-JQ1 treatment and vehicle control was statistically significant (uncorrected $p=0.024$, corrected $p=0.047$); however, the planned comparison between (+)-JQ1 and (−)-JQ1 treatment was not statistically significant (uncorrected $p=0.093$, corrected $p=0.187$). Interestingly, this result suggests the (−)-JQ1 enantiomer might possess pharmacological activity even though relative *MYC* expression remained largely unchanged following treatment *in vitro* (*Figure 1*). This could be due to how (−)-JQ1 interacts with and is metabolized with other molecules *in vivo* compared to (+)-JQ1, thus leading to unexpected effects (*Chhabra et al., 2013*; *Nguyen et al., 2006*).

In addition to monitoring overall survival, we assessed tumor burden, as measured by bioluminescence imaging, during the first 22 days of treatment (*Figure 3*). Mice treated with vehicle control or (+)-JQ1 on day 22 reached an average bioluminescence of $2.73 \times 10^8$ [n = 7, *SD* = $6.01 \times 10^8$] and

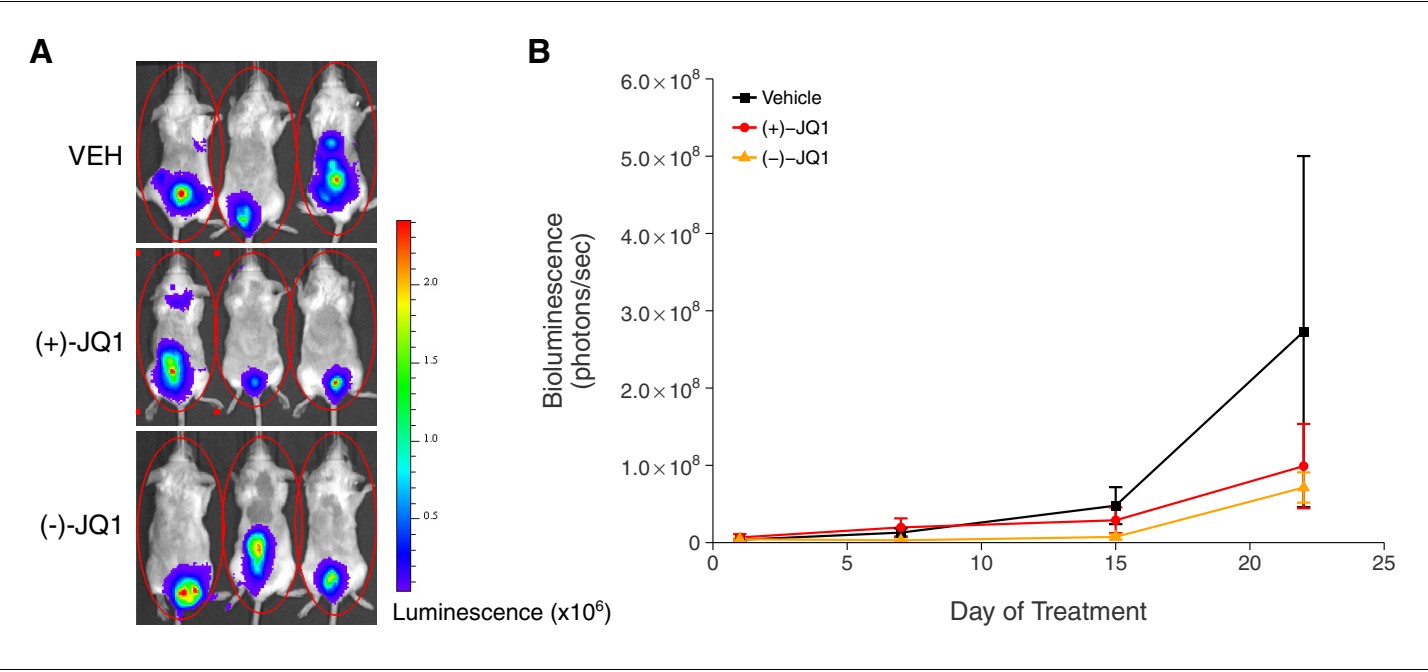

**Figure 3.** Tumor burden in JQ1-treated MM.1S-luc orthotopic xenograft model. Female SCID-beige mice were orthotopically xenografted after intravenous injection with MM.1S-luc cells. Following detection of established disease (diffusely engrafted in the skeleton with an increase in bioluminescence), mice were randomly assigned to receive daily IP injections of 50 mg/kg (+)-JQ1, 50 mg/kg (−)-JQ1, or vehicle control (VEH). (A) Representative whole-body bioluminescence images of mice bearing MM.1S tumors 22 days after the start of the indicated treatment. (B) Line graph of tumor burden, as measured by whole-body bioluminescence, of tumor bearing mice during the course of the indicated treatment. Means reported and error bars represent s.e.m. The number of mice per condition at start of treatment (n = 12 for each group) and at day 22 ((+)-JQ1 = 11, (−)-JQ1 = 9, Vehicle = 7). One-way ANOVA on a natural log transformed day 22 bioluminescence signal; $F_{(2,24)} = 1.126$, $p=0.341$. The pairwise contrast between (+)-JQ1 treatment and Vehicle; Fisher's LSD test; $t(24) = 1.303$, $p=0.205$ with *a priori* alpha level = 0.05. The pairwise contrast between (+)-JQ1 and (−)-JQ1 treatment; Fisher's LSD test; $t(24) = 1.221$, $p=0.234$ with *a priori* alpha level = 0.05. Additional details for this experiment can be found at https://osf.io/pnvtd/.

The following figure supplement is available for figure 3:

**Figure supplement 1.** Individual tumor xenografts.

$9.90\times10^7$ [n = 11, $SD = 1.81\times10^8$], respectively. This compares to an average bioluminescence of $1.85\times10^{10}$ [n = 10, $SD = 1.01\times10^{10}$] in vehicle control treated mice and $5.52\times10^9$ [n = 9, $SD = 2.25\times10^9$] in (+)-JQ1 treated mice reported in the original study (*Delmore et al., 2011*). Finally, tumor burden at day 22 in mice treated with (−)-JQ1 reached an average bioluminescence of $7.13\times10^7$ [n = 9, $SD = 5.87\times10^7$]. To test if (+)-JQ1 treatment decreased the burden of disease, we performed a one-way ANOVA on the day 22 bioluminescence signal (natural log-transformed), which was not statistically significant, $F_{(2,24)} = 1.126$, $p=0.341$. Additionally, as outlined in the Registered Report (*Kandela et al., 2015*), we planned to conduct two comparisons using Fisher's least significant difference (LSD) test. The planned comparison of tumor burden from mice treated with (+)-JQ1 compared to vehicle control was not statistically significant ($t(24) = 1.303$, $p=0.205$), nor was the planned comparison between (+)-JQ1 treatment and (−)-JQ1 ($t(24) = 1.221$, $p=0.234$). These results should take into consideration that some mice were euthanized due to disease progression before the prespecified tumor burden analysis endpoint of 22 days (*Figure 3—figure supplement 1*), which did not occur in the original study. This led to a sample size imbalance due to the preferential death of mice in the vehicle control group (5 of 12) and (-)-JQ1 group (3 of 12) compared to the (+)-JQ1 group (1 of 12). The higher rate of early death in the vehicle control and (-)-JQ1 groups is reflective of the increased overall survival observed in (+)-JQ1 treated mice (*Figure 2*). Besides leading to information loss, if the mice that were euthanized ended up having high tumor burdens, then the group averages reported in this replication attempt would be distorted, limiting the opportunity to

detect a statistically significant difference (*Laajala et al., 2012*). This could be due to biological variability of the kinetics of engraftment and growth of systemically injected tumor cells, as well as differences in the details between the original experimental design and this replication attempt that were not known or accounted for.

There are a number of factors that can affect the assessment of tumor growth *in vivo* using bioluminescence imaging. The depth and location of the tumor as well as the thickness or color of the animal's skin can alter the bioluminescent signal (*Baba et al., 2007*), as well as the choice of anesthesia used (*Keyaerts et al., 2012*). The route of administration and the timing of imaging after D-luciferin injections can also impact the signal. While this study injected D-luciferin by intraperitoneal injection, same as the original study, this route of injection can lead to variations in the signal due to variations in the rate of absorption across the peritoneum to reach the luciferase expressing cells (*Close et al., 2011*). This makes intravenous (*Keyaerts et al., 2008*) or subcutaneous (*Khalil et al., 2013*) administration of D-luciferin attractive alternatives. Further, the signal, and thus the perception of tumor burden, can also vary during longitudinal monitoring since the time course of the bioluminescence signal after D-luciferin injection changes with days after the luciferase expressing cells were inoculation (*Inoue et al., 2010*).

## Meta-analyses of original and replicated effects

We performed a meta-analysis using a random-effects model to combine each of the effects described above as pre-specified in the confirmatory analysis plan (*Kandela et al., 2015*). To provide a standardized measure of the effect, a common effect size was calculated for each effect from the original and replication studies. Cohen's $dz$ is the standardized difference between the two correlated measurements using the standard deviation of the difference scores. While, Cohen's $d$ is the standardized difference between two independent means using the pooled sample standard deviation. The hazard ratio (HR) is the ratio of the probability of a particular event, in this case death, in one group compared to the probability in another group.

There were two comparisons made with the qRT-PCR data evaluating *MYC* expression in (+)-JQ1 treated MM.1S-luc cells (*Figure 4A*). The comparison between (+)-JQ1 at 1 hr to (+)-JQ1 at 0 hr resulted in $dz = 17.41$, 95% CI [6.02, 28.41] in this study, which compares to $dz = 2.48$, 95% CI [−0.59, 5.79] for the data estimated a priori from *Figure 3B* in the original study assuming two

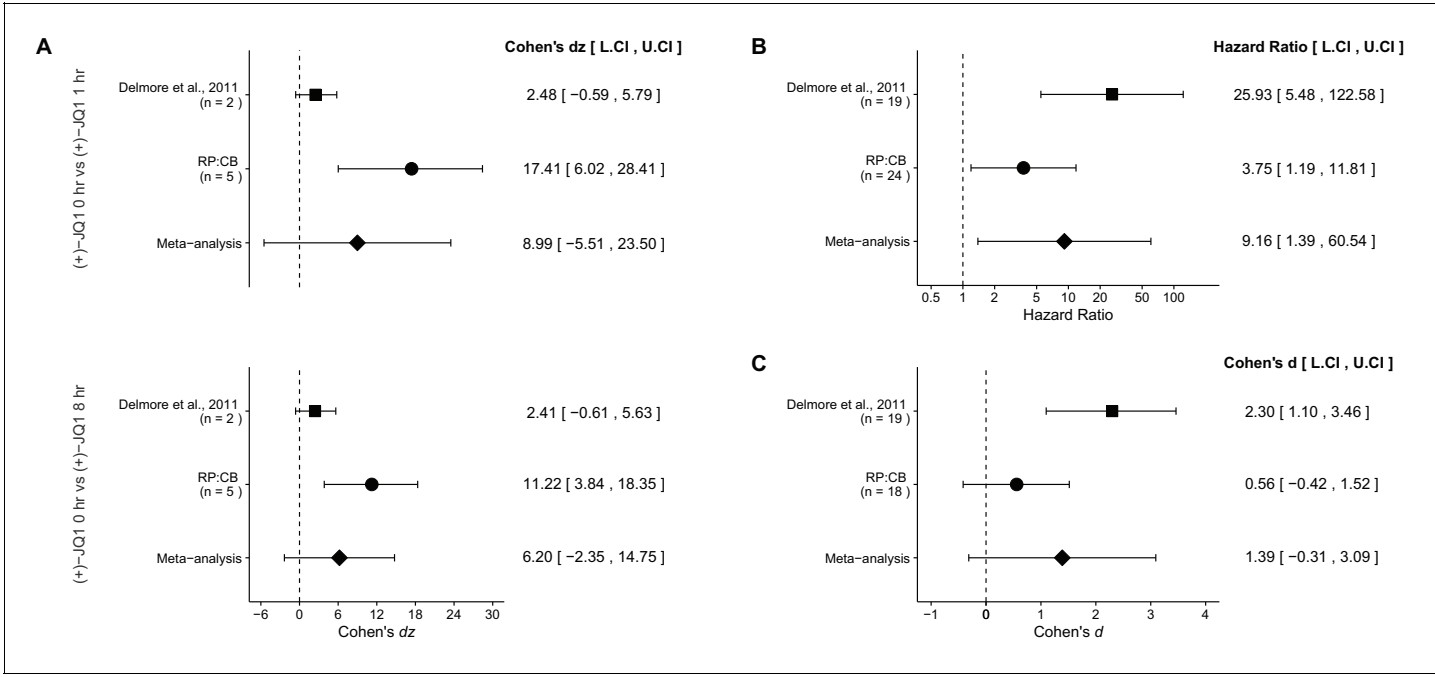

**Figure 4.** Meta-analyses of each effect.

biological repeats as stated in the methods (*Delmore et al., 2011*). A meta-analysis of these two effects resulted in $dz$ = 8.99, 95% CI [−5.51, 23.50], which was not statistically significant, ($p$=0.224). The comparison between (+)-JQ1 at 8 hr to (+)-JQ1 at 0 hr resulted in $dz$ = 11.22, 95% CI [3.84, 18.35] in this study, while the original study was $dz$ = 2.41, 95% CI [−0.61, 5.63], assuming two biological repeats (*Delmore et al., 2011*). A meta-analysis of these two effects resulted in $dz$ = 6.20, 95% CI [−2.35, 14.75], which was not statistically significant ($p$=0.155). For both comparisons the original and replication results are consistent when considering the direction of the effect, however the point estimates of the replication effect sizes were not within the confidence intervals of the original results, or vice versa. Further, the Cochran's $Q$ test for heterogeneity was statistically significant in both meta-analyses (0 hr vs 1 hr, $p$=0.0089; 0 hr vs 8 hr, $p$=0.022), which along with a large confidence intervals around the weighted average effect sizes from the meta-analyses suggests heterogeneity between the original and replication studies. The calculation of the confidence intervals of the above study effect sizes utilize the number of biological repeats as the sample size and not the number of technical replicates. This is because while technical replicates are necessary to evaluate reliability of an assay, it is not independently repeated data and should not be used to properly test scientific hypotheses (*Vaux et al., 2012*).

A comparison of the overall survival distributions for mice treated with (+)-JQ1 or vehicle control resulted in a HR of 3.75, 95% CI [1.19, 11.81] for this study, compared to the original study which resulted in a calculated HR of 25.93, 95% CI [5.48, 122.58]. A meta-analysis of these two effects resulted in a HR of 9.16, 95% CI [1.39, 60.54] was statistically significant ($p$=0.0215). This indicates that the null hypothesis, that the overall survival between mice treated with (+)-JQ1 or vehicle control are not different, can be rejected (*Figure 4B*). Both results are consistent when considering the direction of the effect, however the point estimate of the replication effect size was not within the confidence interval of the original result, or vice versa.

The comparison of tumor burden, as determined by bioluminescence at day 22, between (+)-JQ1 or vehicle control treated mice bearing MM.1S-luc tumors resulted in $d$ = 0.56, 95% CI [−0.42, 1.52] in this study, which can be compared to $d$ = 2.30, 95% CI [1.10, 3.46] calculated from the data reported in Figure 7D of the original study (*Delmore et al., 2011*). A meta-analysis of these effects resulted in $d$ = 1.39, 95% CI [−0.31, 3.09], which was not statistically significant ($p$=0.110) (*Figure 4C*). Both results are consistent when considering the direction of the effect, however the point estimate of the replication effect size was not within the confidence interval of the original result, or vice versa. The large confidence intervals of the meta-analysis along with a statistically significant Cochran's $Q$ test ($p$=0.035), suggests heterogeneity between the original and replication studies.

This direct replication provides an opportunity to understand the present evidence of these effects. Any known differences, including reagents and protocol differences, were identified prior to conducting the experimental work and described in the Registered Report (*Kandela et al., 2015*). However, this is limited to what was obtainable from the original paper and through communication with the original authors, which means there might be particular features of the original experimental protocol that could be critical, but unidentified. So while some aspects, such as cell line, strain and sex of mice, number of cells injected, and the drug dose was maintained, others were unknown or not easily controlled for. These include variables such as cell line genetic drift (*Hughes et al., 2007*; *Kleensang et al., 2016*), as well as subclonal drift in heterogeneous stable cells (*Shearer and Saunders, 2015*), circadian biological responses to therapy (*Fu and Kettner, 2013*), the microbiome of recipient mice (*Macpherson and McCoy, 2015*), housing temperature in mouse facilities (*Kokolus et al., 2013*), and differing compound potency resulting from different stock solutions (*Kannt and Wieland, 2016*). Whether these or other factors influence the outcomes of this study is open to hypothesizing and further investigation, which is facilitated by direct replications and transparent reporting.

## Materials and methods

As described in the Registered Report (*Kandela et al., 2015*), we attempted a replication of the experiments reported in Figures 3B and 7C–E of (*Delmore et al., 2011*). A detailed description of all protocols can be found in the Registered Report (*Kandela et al., 2015*). Additional detailed

experimental notes, data, and analysis are available on the Open Science Framework (OSF) (RRID: SCR_003238) (https://osf.io/7zqxp/; *Aird et al., 2016*).

## Cell culture

MM.1S-luc cells (Dr. Andrew Kung, Columbia University Medical Center), expressing luciferase and neomycin after retroviral transduction with a pMMP-LucNeo vector and G418-selection, were previously described (*Mitsiades et al., 2004*). Cells were maintained in RPMI 1640 with 2 mM L-glutamine (Life Technologies, cat# 11875–093) supplemented with 10% Fetal Bovine Serum (FBS) (Life Technologies, cat# 16000–044), 100 U/ml penicillin, 50 $\mu$g/ml streptomycin, and 200 $\mu$g/ml G-418 sulfate (Life Technologies, cat# 10131–035) at 37°C in a humidified atmosphere at 5% $CO_2$. Quality control data for the MM.1S cell line is available at https://osf.io/r38p3/. This includes results confirming the cell line was free of mycoplasma contamination and common mouse pathogens. Additionally, STR DNA profiling of the cell line was performed and cells were confirmed to be the indicated cell line when queried against STR profile databases.

## Therapeutic compounds

5 mg each of (+)-JQ1 and (−)-JQ1 (JQ1 enantiomer set, EMD Millipore, cat# 500586; *in vitro* studies, (+)-JQ1 enantiomer lot# D00147531, (−)-JQ1 enantiomer lot# D00147532; *in vivo* studies, (+)-JQ1 enantiomer lot# 2630921, (−)-JQ1 enantiomer lot# 2639083) was dissolved in 1.094 ml DMSO (ATCC, 4-x-5) to generate a 10 mM stock, which was filter sterilized, aliquoted, and stored at −20°C until use for *in vitro* experiments. For *in vivo* applications, one vial (5 mg) each of (+)-JQ1 and (−)-JQ1 was dissolved in 1 ml of 5% (w/v) dextrose in sterile water as pre-specified in the Registered Report, but after extensive vortexing, the solution remained milky unless filter sterilized through a 0.22 $\mu$m filter. The formulation was changed to diluting 5 mg of each compound in 80 $\mu$l of DMSO (Sigma-Aldrich, cat# D8418) and then further diluting to 1 ml by addition of 920 $\mu$l of 10% (w/v) 2-Hydroxypropyl-$\beta$-cyclodextrin (HP-$\beta$-CD) (Sigma-Aldrich, cat# H107-5G) dropwise while vortexing to give a final concentration of 5 mg/ml. Additional details are available at: https://osf.io/kwu3f/. The final concentration of DMSO was 8% (v/v). The vehicle was 10% HP-$\beta$-CD with 8% DMSO (v/v). Aliquots were stored at −20°C until use with no repeated cycles of freeze-thaws.

## Gene expression analysis

Cells were seeded at $8\times10^5$ cells per 35 mm tissue culture dish in 1.8 ml growth medium. The next day 10 mM stock solutions of (+)-JQ1 and (−)-JQ1 were diluted 1:2000 in growth medium to 5 $\mu$M and 0.2 ml was added to medium in each dish to give a final concentration of 500 nM. For vehicle control, DMSO was diluted 1:2000 in growth medium (0.05% DMSO). After 0, 1, or 8 hr, cells were centrifuged and each plate was lysed in 200 $\mu$l TRI Reagent following manufacturer's instructions. Purified RNA was dissolved in 20 $\mu$l RNase-free water and RNA concentration and purity was determined using an Agilent 2100 Bioanalyzer (quality control data available at https://osf.io/d2a5w/). Total RNA was reverse transcribed into cDNA using a First-Strand cDNA Synthesis Kit (GE Healthcare, cat# GE27-9261-01) following manufacturer's instructions. Reaction volume was 21 $\mu$l and consisted of 7 $\mu$l first-strand reaction mix, 1 $\mu$l random hexamers (0.2 $\mu$g pd[N]6), 1 $\mu$l DTT (200 mM), and 12 $\mu$l RNA diluted in water. An initial attempt with 1 $\mu$g total RNA was tried, but the qRT-PCR results indicated that this was too high, and was reduced to 0.1 $\mu$g total RNA for all subsequent assays. A negative control contained 12 $\mu$l of water with no RNA template was also included. qRT-PCR was performed using a real-time PCR system (Applied Biosystems, cat# 7900HT). Reaction volumes were 20 $\mu$l and consisted of 10 $\mu$l 2X TaqMan Gene Expression Master Mix (Life Technologies, cat# 4369016), 1 $\mu$l 20X TaqMan Gene Expression Assay (Applied Biosystems, cat# 4331182) for either *MYC* (Hs00905030_m1) or *GAPDH* (Hs02758991_g1), 2 $\mu$l cDNA template, and 7 $\mu$l water. A negative control contained 9 $\mu$l water with no cDNA template was also included. Reactions were performed in technical triplicate with the following parameters: 2 min/50°C, 10 min/95°C, and 40 cycles of 15 s/ 95°C and 1 min/60°C. *MYC* transcript levels were normalized to *GAPDH* levels in each sample using the $\Delta\Delta$Ct method.

## Animals

All animal procedures were approved by the Northwestern University IACUC# IS00000556 and were in accordance with the Northwestern University's policies on the care, welfare, and treatment of laboratory animals. No blinding occurred during the experiments.

Six-week old female SCID-beige mice (Charles River, strain code 250) were housed in cages up to five mice and offered Certified Rodent Diet (Harlan Teklad, cat # 7912) *ad libitum*. The animal room was set to maintain between 68–75 °F, a relative humidity of 30–70%, a minimum of 15 room air changes per hour, and a 12 hr light/dark cycle, which was interrupted for study-related activities.

Animals were checked twice within the first day after MM.1S tumor cell inoculation for mortality, abnormalities, and signs of pain or distress. Cage-side observations were done daily. Body weight (scale: Scout Pro O'Haus, Parsippany, New Jersey; model # SP202) was recorded weekly after inoculation and daily after JQ1 treatments.

## MM.1S inoculation

After one week of acclimation, female SCID-beige mice were inoculated intravenously (IV) via the lateral tail vein with $2\times10^6$ MM.1S-luc cells suspended in 200 µl Phosphate Buffered Saline (PBS) (Sigma-Aldrich, cat# D8537). For inoculation, cells were washed 2X with PBS and resuspended in PBS at $10^7$ cells/ml. Immediately prior to inoculation, D-luciferin (Promega, cat# P1042) was added to a separate dish of cells in culture and imaged in a Xenogen IVIS Spectrum (Caliper Life Sciences, Hopkinton, Massachusetts) to confirm expression of the luciferase construct (image available at https://osf.io/4a2jw/).

A pilot study was conducted to establish the timescale of tumor progression that would achieve a similar tumor burden, as measured by whole-body bioluminescence ($\sim2\times10^6$), as observed in the original study after an initial attempt to engraft mice was unsuccessful in detecting bioluminescence even though disease progression (hind limb paralysis) was observed. For the pilot study, five mice were inoculated with MM.1S cells. Eight days after inoculation mice were injected intraperitoneally (IP) with 75 mg/kg of D-luciferin in 150 $\mu$l PBS and imaged 15 min later under isoflurane anesthesia in a Xenogen IVIS Spectrum. To monitor tumor progression, imaging was repeated at day 13 and 15 after inoculation and thereafter three times a week until day 57. The target bioluminescence of $\sim2\times10^6$ was achieved at day 27 after inoculation. Pilot study and initial attempt data available at https://osf.io/jse9q/.

A total of thirty-eight female SCID-beige mice were inoculated with MM.1S cells with the aim of randomizing mice into different treatment groups after the target bioluminescence of $\sim2\times10^6$ was achieved. Fourteen days after inoculation mice were injected intraperitoneally (IP) with 75 mg/kg of D-luciferin in 150 $\mu$l PBS and imaged 30 min later under isoflurane anesthesia in a Xenogen IVIS Spectrum. To monitor tumor progression, imaging was repeated at day 21, 27, and 30 after inoculation until the mean bioluminescence reached the target of $\sim2\times10^6$, which was observed on day 30. Similar to the original study, imaging was repeated five days later and the difference in bioluminescence (day 35 minus day 30) in each mouse was calculated. Thirty-six mice showed established disease (lesions diffusely engrafted in the skeleton with an increase in bioluminescence between day 30 and 35) and were randomized into three groups using the alternating serpentine method. Randomization was consistent with even distribution across all groups (one-way ANOVA: $F(2,33) = 0.027$, $p=0.973$). Mice were injected daily with (+)-JQ1, (−)-JQ1, or vehicle control for 36 days. Tumor progression was monitored at day 1, 7, 22, and 29 of treatment, and survival was monitored until day 36.

## JQ1 administration

Mice were weighed daily and injected IP daily with (+)-JQ1 or (−)-JQ1 (5 mg/ml) in a volume of 10 $\mu$l/g body weight to achieve a final dose of 50 mg/kg. Vehicle (10% HP-$\beta$-CD, 8% DMSO) was injected IP in a volume of 10 $\mu$l/g body weight.

## IVIS imaging

Images were acquired in a Xenogen IVIS Spectrum at a medium binning level (8) and a 22.6 cm field of view. Acquisition times were set to auto-exposure, and ranged from 60 to 0.5 s depending on the intensity of luminescence. Living Image software (RRID:SCR_014247), version 4.3, was used for

quantitative analysis of the bioluminescence signal. Image analysis files are available at https://osf.io/jse9q/.

## Statistical analysis

Statistical analysis was performed with R software (RRID:SCR_001905), version 3.2.3 ( *R Core Team, 2016*). All data, csv files, and analysis scripts are available on the OSF (https://osf.io/7zqxp/). Confirmatory statistical analysis was pre-registered (https://osf.io/ng3st/) before the experimental work began as outlined in the Registered Report (*Kandela et al., 2015*). Data were checked to ensure assumptions of statistical tests were met. When described in the results, the Bonferroni correction, to account for multiple testings, was applied to the alpha error or the *p*-value. The Bonferroni corrected value was determined by dividing the uncorrected value by the number of tests performed. For the alpha error this resulted in an *a priori* significance threshold of 0.025 (0.05/2). Although the Bonferroni method is conservative, it was accounted for in the power calculations to ensure sample size was sufficient. In cases where the number of groups were three and the sample sizes were evenly distributed among the groups, Fisher's LSD test was performed resulting in an *a priori* significance threshold of 0.05. For the qRT-PCR data analysis, we performed a mixed-design analysis of variance (ANOVA) on normalized *MYC* expression with time (0 hr, 1 hr, and 8 hr) as the within-subjects factor and treatment ((+)-JQ1, (−)-JQ1, or vehicle) as the between-subjects factor with an *a priori* significance threshold of 0.05. A meta-analysis of a common original and replication effect size was performed with a random effects model and the *metafor* R package (*Viechtbauer, 2010*) (available at https://osf.io/9snja/). The original study data presented in *Figure 3B* was extracted *a priori* from the published figure by determining the mean and upper/lower error values for each data point, while the raw data pertaining to Figure 7D,E were shared by the original authors. The summary data was published in the Registered Report (*Kandela et al., 2015*) and was used in the power calculations to determine the sample size for this study.

## Deviations from registered report

The vehicle formulation for *in vivo* applications was changed from what is listed in the Registered Report. The Registered Report listed it as 5% dextrose in water (D5W) while the replication experiment used 10% HP-$\beta$-CD with 8% DMSO (v/v). The vehicle listed in the original study was indicated as D5W, however 10% HP-$\beta$-CD, 8% DMSO is described here as a vehicle which improves the solubility of the compound. Additional materials and instrumentation not listed in the Registered Report, but needed during experimentation are also listed.

An initial attempt to inoculate 33 animals with MM.1S-luc cells as outlined in the Registered Report was unsuccessful in detecting bioluminescence even though disease progression (hind limb paralysis) was observed and the MM.1S-luc cells had a strong luminescent signal prior to injection. MM.1S-luc cells were grown under antibiotic selection to enrich for luciferase expressing cells and a pilot study was designed to closely monitor tumor progression and establish the timescale required to achieve a similar tumor burden as that observed in the original study. Following this modification, which is reported here, the number of mice to inoculate was increased to 38 and monitored until the desired bioluminescence was achieved, which occurred on day 30 post-inoculation, after which the remaining aspects of the protocol were carried out.

## Acknowledgements

The Reproducibility Project: Cancer Biology would like to thank Dr. Andrew Kung (Memorial Sloan Kettering Cancer Center) for sharing critical reagents and data, specifically the MM.1S-luc cells. We would also like to thank the following companies for generously donating reagents to the Reproducibility Project: Cancer Biology; American Type and Tissue Collection (ATCC), Applied Biological Materials, BioLegend, Charles River Laboratories, Corning Incorporated, DDC Medical, EMD Millipore, Harlan Laboratories, LI-COR Biosciences, Mirus Bio, Novus Biologicals, Sigma-Aldrich, and System Biosciences (SBI).

# Additional information

### Group author details

Reproducibility Project: Cancer Biology

Elizabeth Iorns: Science Exchange, Palo Alto, United States; Alexandria Denis: Center for Open Science, Charlottesville, Virginia; Stephen R Williams: Center for Open Science, Charlottesville, Virginia; Nicole Perfito: Science Exchange, Palo Alto, California; Timothy M Errington, http://orcid.org/0000-0002-4959-5143: Center for Open Science, Charlottesville, Virginia

### Competing interests

FA, IK, CM: Developmental Therapeutics Core is a Science Exchange associated lab. RP:CB: EI, NP: Employed by and hold shares in Science Exchange Inc. The other authors declare that no competing interests exist.

### Funding

| Funder | Author |
|---|---|
| Laura and John Arnold Foundation | Reproducibility Project: Cancer Biology |

The funder had no role in study design, data collection and interpretation, or the decision to submit the work for publication.

### Author contributions

FA, IK, CM, Acquisition of data, Drafting or revising the article; RP:CB, Analysis and interpretation of data, Drafting or revising the article

### Ethics

Animal experimentation: All animal procedures were approved by the Northwestern University IACUC# IS00000556 and were in accordance with the Northwestern University's policies on the care, welfare, and treatment of laboratory animals.

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
