## [Decision Letter]

Thank you for resubmitting your work entitled "Replication Study: BET Bromodomain Inhibition as a Therapeutic Strategy to Target c-Myc" for further consideration at *eLife*. Your article has been favorably evaluated by Charles Sawyers (Senior Editor) and two reviewers, one of whom is a Reviewing Editor.

The replication study of " BET Bromodomain Inhibition as a Therapeutic Strategy to Target c-Myc " is proposed to replicate the key findings in the original paper published in Cell by Delmore et al. in 2011. The experiments set up in this replication study are consistent with their published registered report in *eLife* last year. The description of the experiments is very detailed and the mathematical statistical methods used in this replication study are objective, proper and accurate. There are some minor issues that need to be addressed before acceptance, as outlined below:

1) It is recommended that the lengthy section of statistical methodology described in the Results section be distilled and the majority of this information moved to the Materials and methods section. This will make the manuscript more approachable by the general reader.

2) It looks like (-)-JQ1 also has effects on overall survival and tumor burden mentioned in Figure 2 and Figure 3, even though it doesn't affect the mRNA level of cMyc (Figure 1). The authors may wish to provide some explanation.

3) In order to interpret why they did not observe the baseline after inoculation (~2x106 bioluminescence) until day 27 post-inoculation in the pilot study (The original study observed it 8 days post-inoculation!), the authors wrote a paragraph: "Two studies did not report baseline signals until 2 weeks after inoculation, granted these studies injected 5x106 cells opposed to 2x106 cells injected in the original study and this replication attempt (Mitsiades et al., 2004; Tai et al., 2014). Similarly, Zhang and colleagues did not observe this level of baseline bioluminescence until 2-3 weeks when injecting 3x106 cells (Zhang et al., 2014). Additionally, Azab and colleagues, which inoculated 2x106 MM.1S-luc cells, reported that it took 3-4 weeks for sufficient tumor progression to occur and be detected (Azab et al., 2009)."

It is true that the two papers of Tai et al. and Zhang et al. used almost the same in vivo condition (gender, age and injection mode) compared with the original paper. However, the mice used in the paper of Mitsiades et al. were 6- to 8-week-old male SCID/NOD mice, not female SCID-beige mice; also, the mice used in the paper of Azab et al. are male, 7 to 9 weeks old, severe combined immunodeficient (SCID) mice. Thus, it is obviously not true that the observations from these two papers of Mitsiades et al. and Azab et al. can explain the variation of the bioluminescence baseline because of the different experimental systems. This should be mentioned.

4) The planned comparison of tumor burden from mice treated with (+)-JQ1 compared to vehicle control was not statistically significant, nor was the planned comparison between (+)-JQ1 treatment and (-)-JQ1. The reason could be because some mice were euthanized due to disease progression before the pre-specified tumor burden analysis endpoint of 22 days. If the same phenomenon did not occur in the original paper, there might exist some different details between this replication study and the original paper. This should be discussed.

5) The statistical method in the original paper of Figure 3 is "paired Student's t test each relative to t = 0 hr)" described in the figure legend of Figure 3 of the original paper. However, the authors depicted it as unpaired in the third paragraph of the subsection “Evaluation of MYC expression in JQ1-treated MM.1S-luc cells” in this replication study. This should be noted.

---

## [Author Response]

*The replication study of " BET Bromodomain Inhibition as a Therapeutic Strategy to Target c-Myc " is proposed to replicate the key findings in the original paper published in Cell by Delmore et al. in 2011. The experiments set up in this replication study are consistent with their published registered report in eLife last year. The description of the experiments is very detailed and the mathematical statistical methods used in this replication study are objective, proper and accurate. There are some minor issues that need to be addressed before acceptance, as outlined below:*

*1) It is recommended that the lengthy section of statistical methodology described in the Results section be distilled and the majority of this information moved to the Materials and methods section. This will make the manuscript more approachable by the general reader.*

We have consolidated some of the statistical methodology in the Results/Discussion section in the revised manuscript. The revisions pertained mostly with the first section describing the results from the *MYC* expression experiment. However, we’ve still maintained the descriptive statistics and most of the inferential statistics as they provide one of the means for a reader to infer the outcome of the replication attempt.

*2) It looks like (-)-JQ1 also has effects on overall survival and tumor burden mentioned in Figure 2 and Figure 3, even though it doesn't affect the mRNA level of cMyc (Figure 1). The authors may wish to provide some explanation.*

We have added adding some additional discussion on this observation and how this could be due to how the (-)-JQ1 enantiomer might be processed in vivo leading to unexpected effects compared to (+)-JQ1.

*3) In order to interpret why they did not observe the baseline after inoculation (~2x106 bioluminescence) until day 27 post-inoculation in the pilot study (The original study observed it 8 days post-inoculation!), the authors wrote a paragraph: "Two studies did not report baseline signals until 2 weeks after inoculation, granted these studies injected 5x106 cells opposed to 2x106 cells injected in the original study and this replication attempt (Mitsiades et al., 2004; Tai et al., 2014). Similarly, Zhang and colleagues did not observe this level of baseline bioluminescence until 2-3 weeks when injecting 3x106 cells (Zhang et al., 2014). Additionally, Azab and colleagues, which inoculated 2x106 MM.1S-luc cells, reported that it took 3-4 weeks for sufficient tumor progression to occur and be detected (Azab et al., 2009)."*

*It is true that the two papers of Tai et al. and Zhang et al. used almost the same in vivo condition (gender, age and injection mode) compared with the original paper. However, the mice used in the paper of Mitsiades et al. were 6- to 8-week-old male SCID/NOD mice, not female SCID-beige mice; also, the mice used in the paper of Azab et al. are male, 7 to 9 weeks old, severe combined immunodeficient (SCID) mice. Thus, it is obviously not true that the observations from these two papers of Mitsiades et al. and Azab et al. can explain the variation of the bioluminescence baseline because of the different experimental systems. This should be mentioned.*

Thank you for raising this point. We agree on being explicit about how these studies differ and have revised this section of the manuscript to reflect the different strains and sexes utilized in the papers by Mitsiades et al. and Azab et al.

*4) The planned comparison of tumor burden from mice treated with (+)-JQ1 compared to vehicle control was not statistically significant, nor was the planned comparison between (+)-JQ1 treatment and (-)-JQ1. The reason could be because some mice were euthanized due to disease progression before the pre-specified tumor burden analysis endpoint of 22 days. If the same phenomenon did not occur in the original paper, there might exist some different details between this replication study and the original paper. This should be discussed.*

We agree and have highlighted this aspect at the end of the paragraph discussing the result of the tumor burden to provide a stronger transition to the next paragraph that discusses various factors that could have contributed to these differences.

*5) The statistical method in the original paper of Figure 3 is "paired Student's t test each relative to t = 0 hr)" described in the figure legend of Figure 3 of the original paper. However, the authors depicted it as unpaired in the third paragraph of the subsection “Evaluation of MYC expression in JQ1-treated MM.1S-luc cells” in this replication study. This should be noted.*

We performed the same statistical test as reported in the original paper, a paired Student’s t-test, however as described in the Registered Report we also performed the analysis as a between subjects design since the experimental set-up suggests this analysis. In the Replication Study we described the unpaired t-test as an additional exploratory analysis since it was not originally reported (revised text: “Although this was a planned analysis, it is exploratory because it differs from the methods used in the original study by analyzing the data as unpaired instead of paired.”). We have also revised the statistical section for this experiment as suggested in first comment to better clarify this aspect.